# Dietary Net Energy Concentration Affects Growth Performance, Carcass Traits, Intramuscular Fatty Acid Profile, and Cecal Microbiota of Pigs with Restricted Feed Allowance

**DOI:** 10.3390/ani15243514

**Published:** 2025-12-05

**Authors:** Qinfeng Hu, Wanxin Xiang, Youzhi Pu, Yong Zhang, Pan Zhou, Tiande Zou, Zhengjun Xie, Zhiqing Wu, Xiang Ao, Jinming You, Honglin Yan

**Affiliations:** 1College of Life Science and Agri-Forestry, Southwest University of Science and Technology, Mianyang 621010, China; qinfenghu02@163.com (Q.H.); 13438461953@163.com (W.X.); puyouzhi2023@163.com (Y.P.); zyzlrzjh@swust.edu.cn (Y.Z.); pamelazhoupanpan@aliyun.com (P.Z.); 2Jiangxi Province Key Laboratory of Animal Nutrition, College of Animal Science and Technology, Jiangxi Agricultural University, Nanchang 330095, China; tiandezou@jxau.edu.cn; 3Shuangbaotai Group Co., Ltd., Nanchang 330095, China; xiezj@sbtjt.com (Z.X.);; 4Guangnan (Zhanjiang) Jiafeng Feed Co., Ltd., Zhanjiang 524033, China; 5Chengdu Tieqilishi Feed Co., Ltd., Sichuan TQLS Group, Chengdu 610200, China; ao_sunshine@hotmail.com

**Keywords:** intramuscular fat content, gut microbiota, meat quality, lipid metabolism

## Abstract

Adjusting feeding frequency and dietary net energy (NE) concentration is crucial for the efficiency of pig production. As an important local pig breed in China, the Sichuan-Tibetan black pig has significant breeding potential, but its efficient rearing mode still needs further optimization. This study provides practical insights for improving the rearing of Sichuan-Tibetan black pigs. Regulating feeding frequency and dietary energy concentration can effectively reduce costs while enhancing pork production efficiency and improving meat quality. Compared to common commercial pig breeds, Sichuan-Tibetan black pigs exhibit unique characteristics in growth performance, carcass traits, and meat quality. The findings of this study provided a more precise planning basis for their feeding management, thereby reducing environmental impacts and production costs during the rearing process. In addition, under the restricted feeding allowance, the adjustment of diet NE concentration improved the deposition ratio of carcass muscle and fat to better meet consumers’ demand for high-quality pork. It can contribute to the conservation of local pig germplasm resources, thus providing support for promoting the development of sustainable pork production systems.

## 1. Introduction

Altering feeding frequency has emerged as a particularly effective approach towards improved growth traits and meat quality of pigs [1]. However, contradictory findings have been reported regarding the effects of higher or lower feeding frequency on growth-related phenotypes of pigs. Some studies have indicated that increasing feeding frequency could improve nutrient utilization efficiency in swine [2,3], while others have demonstrated that lower feeding frequency might enhance feed efficiency of swine compared to ad libitum [4,5]. Despite this, higher or lower feeding frequency impacted the growth performance of pigs associated with changes in energy metabolism, indicating that altered feeding frequency could decrease fat deposition or promote muscle accretion of pigs.

Dietary energy density as a determinant factor in regulating energy metabolism has been demonstrated to affect the growth-related traits and meat quality in meat-producing animals. The partition of energy into adipose or muscle tissues contributed to the feed efficiency of pigs [6]. Insufficient dietary energy supply led to growth retardation and compromised myogenesis, while energy excess promoted meat quality deterioration in pigs [7,8]. In the practical feeding of Chinese indigenous pig breeds, restricted feeding frequency has been applied to control the fat deposition to achieve a better carcass composition and meat quality with a compromised growth rate. However, existing studies lacked evidences to define how precise dietary energy density adjustments could achieve a balance between growth rate and carcass composition/meat quality in pigs under the condition of restricted feeding frequency. Additionally, the effect of dietary energy density on energy partitioning of pigs that were offered restricted feeding allowance remains unknown.

Gut microbiota has been shown to play a key role in dietary energy-induced alterations of carcass composition in meat-producing animals [9,10]. Imbalanced gut microbiota has been associated with excessive lipid deposition [11]. In addition, certain changes in taxa abundances have been correlated with whole-body fat content [12,13]. However, it is uncertain whether dietary energy level could shape the gut microbiota composition of pigs under the restricted feeding condition.

We hypothesized that the optimal dietary net energy would improve the carcass traits and meat quality of pigs that offered restricted feeding allowance. This study aimed to investigate the influence of dietary energy levels on the growth performance, carcass traits, meat quality, intramuscular lipid metabolism, and cecal microbiota composition of pigs under restricted feeding frequency conditions.

## 2. Materials and Methods

### 2.1. Experiment Design and Animal Management

A total of 32 healthy castrated male Sichuan-Tibetan black pigs with similar initial body weights (25.98 ± 0.27 kg) were randomly allocated into four groups. Each group had eight replicates of one pig per replicate pen. Diets were formulated to meet the feeding standards for Chinese local pig breeds (GB/T 39235-2020) [14]. All diets were iso-nitrogenous, with identical crude protein content (Table 1). Pigs in the CON group were provided with a basal diet (2330 kcal NE kg^−1^) on an ad libitum basis, while pigs in the other three treatment groups received diets formulated to contain 2330, 2370, and 2410 kcal NE kg^−1^, respectively, with twice-daily feeding at 8:00 and 18:00, and each feeding session lasting one hour. The ad libitum pigs had free access to feed throughout the trial. The temperature in the pig environment was maintained at 22–25 °C, and the relative humidity was maintained at 60–65%. In order to calculate average daily gain (ADG), average daily feed intake (ADFI), and feed to gain ratio (F/G), the initial body weight and final body weight were measured, and the consumption amount of feed for each pig was recorded on a weekly basis.

### 2.2. Sample Collection

The experiment lasted for 19 weeks. At the end of the feeding trial, after 8 h of fasting, all the experimental pigs were euthanized via electrical stunning and subsequently slaughtered. After removing the head, hooves, and tail, the carcass weight was measured, followed by splitting the carcass into two half-sides along the midline. The longissimus dorsi muscle (LDM) sample from the left side of the carcass was sealed in a plastic bag and stored at −80 °C for muscle chemical composition and fatty acid analysis. An additional sample of LDM was collected, placed into 1.5 mL EP tubes, and stored at −80 °C for gene expression analysis. LDM samples between the 10th and 13th ribs of the right carcass side were collected for meat quality analysis. The cecum was immediately separated, placed on ice, and punctured using sterilized surgical scissors to extract cecal contents, which were aliquoted into 1.5 mL EP tubes and stored at −80 °C for microbiota composition analysis.

### 2.3. Carcass Traits Measurements

Backfat thickness was determined using a digital caliper in accordance. Triplicate measurements were obtained at three standardized anatomical positions, including the first thoracic vertebra, the last thoracic vertebra, and the last lumbar vertebra on the left side of the carcass. The cross-sectional areas of the LDM were measured using a digital vernier caliper. The loin muscle area was calculated according to the following standardized formula:Loin muscle area=0.7×Loin muscle width×Loin muscle height

### 2.4. Meat Quality Measurements

LDM samples were excised from the 10th–13th rib. Visible connective tissue and surface fat were meticulously removed using sterile surgical instruments, and samples were cut into small pieces and collected to evaluate the meat quality indexes, including the pH, color, cooking loss, and drip loss. Muscle pH was measured using the SFK-Technology pH meter (SFK LEBLANC, Kolding, Denmark). The meat color parameters, including L* (lightness), a* (redness), and b* (yellowness), were measured using the Minolta colorimeter (CR-400, Konica Minolta, Tokyo, Japan). Drip loss was measured as follows: a muscle sample from the 10th rib region was trimmed, weighed, and hung on a metal hook to be suspended in an inflatable plastic bag. After being stored at 4 °C for 48 h, the surface moisture of samples was removed by filter paper blotting prior to reweighing. For cooking loss analysis, LDM samples were vacuum-sealed in plastic bags and immersed in a water bath until the core temperature reached 70 °C. After cooling to ambient temperature, samples were reweighed. Both drip loss and cooking loss were calculated using the following unified formula [15]:Loss(%)=[(Initial weight−Final weight)/Initial weight]×100

### 2.5. Chemical Composition Analysis of Skeletal Muscle

The LDM sample was weighed and placed in a pre-labeled glass dish. Subsequently, it was transferred together with the glass dish into a freeze-dryer for the process of lyophilization under the following conditions: shelf temperature: −50 °C; condenser temperature: −80 °C; chamber pressure: <10 Pa for 48 h until constant weight. After moisture determination, the sample was crushed and sieved through a 40-mesh standard sieve and then was analyzed for EE (method 920.39) and CP (method 954.01) according to AOAC methods [16].

### 2.6. Muscle Fatty Acid Composition Analysis

The analysis of fatty acid (FA) composition in LDM was carried out using a gas chromatograph (Agilent 7890A, Agilent Technologies, Santa Clara, CA, USA) [17]. Initially, total lipids were extracted from the LDM. After homogenizing, the hexane layer was aspirated via anhydrous sodium sulfate for FA analysis. The temperature program was set as follows: the initial column temperature was maintained at 140 °C for 15 min, then increased at a rate of 3 °C per minute until reaching 240 °C, where it was held for another 15 min. The injector and detector temperatures were both set at 250 °C, while the inlet temperature was 220 °C. The injection volume was 1 μL, with a split ratio of 10:1. The gas flow rates were as follows: hydrogen at 30 mL/min, air at 400 mL/min, nitrogen at 40 mL/min, and the carrier gas at 0.8 mL/min. Individual FA peaks were identified by comparing their retention times with those of known standards (Sigma, Tokyo, Japan).

### 2.7. Real-Time Quantitative PCR (RT-qPCR)

The total RNA of the LDM was extracted using RNAiso Plus reagent (Takara, Beijing, China), and the concentration and purity of the total RNA were determined using the Yoke N6000 UV spectrophotometer (YOKE, Tokyo, Japan). Further reverse transcription of cDNA was performed using the Prime Script RT Reverse Transcription Kit (Takara, Chengdu, China). RT-qPCR was conducted using the CFX96 Touch Real-Time PCR Detection System (Bio-Rad Laboratories, Inc., Hercules, CA, USA). Each reaction mixture consisted of 5 μL TB Green Premix Ex Taq II, 0.5 μL forward primer (10 μM), 0.5 μL reverse primer (10 μM), 2 μL sterile water, and 2 μL cDNA template, with a total reaction volume of 10 μL. The thermal cycling conditions were as follows: initial denaturation and enzyme activation at 95 °C for 3 min, followed by 40 cycles of denaturation/annealing/extension and data acquisition (95 °C for 30 s, annealing for 40 s at primer-specific temperatures), and a melt curve analysis from 65 °C to 90 °C with 0.5 °C increments every 5 s. The *ACTB* gene was used as an internal reference for normalization, and relative gene expression levels were calculated using the 2^−ΔΔCt^ method (Table 2) [18].

### 2.8. Cecal Microbiota Analysis

Genomic DNA was extracted from cecal contents using the QiaAmp DNA Stool Mini Kit (Qiagen, Beijing, China), with DNA quality subsequently verified through 1.5% agarose gel electrophoresis. The V3-V4 hypervariable region of the 16S rRNA gene was amplified via PCR using the forward primer (5′-GTGCCAGCMGCCGCGGTAA-3′) and the reverse primer (5′-CCGTCAATTCMTTTRAGTTT-3′). Amplicon libraries were prepared using the Ovation Rapid DR Multiplex System 1-96 (NuGEN, San Carlos, CA, USA), followed by paired-end sequencing on the Illumina MiSeq platform (Illumina, San Diego, CA, USA) for 16S rRNA gene analysis. Raw sequence data were processed and analyzed using Mothur version 1.48.0 to generate amplicon sequence variants (ASV) abundance and taxonomic classification tables. To account for variations in sequencing depth across samples, sequence reads were normalized to the minimum sequencing depth through random subsampling. Subsequent microbial community analyses were performed using R Studio version 3.4.1 with the vegan and phyloseq packages for community ecology analyses.

### 2.9. Statistical Analyses

The results are presented as means ± standard errors of the mean (SEM). All experimental data were analyzed using ANOVA in SAS 9.4 statistical software. Linear and quadratic effects within the restricted feeding frequency group were evaluated using orthogonal polynomials comparison. The *p*-value less than 0.05 were considered statistically different.

## 3. Results

### 3.1. Growth Performance

Compared to the CON group, the final body weight of pigs in NE 2330 group and the ADG of pigs in both the NE 2330 and NE 2370 groups were significantly decreased (*p* < 0.05). The ADFI and F/G of pigs with restricted feeding allowance were significantly lower than those of the CON group pigs (*p* < 0.05). Under the condition of restricted feeding, dietary energy density linearly increased the final body weight, ADG, and ADFI of the pigs and linearly decreased the F/G of the pigs (*p* < 0.05). The ADFI increased in a quadratic (*p* < 0.05) manner as dietary energy density increased, and the F/G decreased quadratically (*p* < 0.05) (Table 3).

### 3.2. Carcass Traits

First rib backfat thickness, lumbosacral junction backfat thickness, and mean backfat thickness of pigs with restricted feeding allowance were significantly lower than those of the CON group pigs (*p* < 0.05) (Table 4). Compared to the CON group, the carcass weight and last-rib backfat thickness of pigs in both the NE 2330 and NE 2370 groups were significantly decreased (*p* < 0.05). Under the condition of restricted feeding, dietary energy density linearly increased all the carcass trait indicators of pigs (*p* < 0.05).

### 3.3. Meat Quality

There were significant differences in the meat quality of pigs across four groups (Table 5). The L*_45min_ of LDM in both NE 2330 and NE 2410 groups and the b*_45min_ of LDM in the NE 2330 group were significantly decreased compared to those in the CON group (*p* < 0.05). With the increase in energy density, the pH_45min_ significantly increased and the b*_45min_ significantly decreased in a quadratic manner (*p* < 0.05). However, there was no significant difference in pH_45min_, a*_45min_, drip loss, and cooking loss among the four groups (*p* > 0.05).

### 3.4. Muscle Chemical Composition Analysis

The EE content in the LDM of the CON group pigs was significantly higher than that of the NE 2330 group pigs (*p* < 0.05) (Table 6). There was no significant difference in the moisture and CP contents of the LDM sample among the four groups (*p* > 0.05).

### 3.5. Fatty Acid Composition

The individual fatty acid concentration in LDM was expressed as the percentage proportion related to the total FA content. Among the four groups, the predominant FAs in LDM were C16:0, C16:1, C18:0, C18:1n9c, and C18:2n6c, which accounted for 90% of all the FAs (Table 7). There was no significant difference in concentrations of all the fatty acids among the four groups (*p* > 0.05).

### 3.6. mRNA Expression of Myosin-Heavy Chain Genes

There were significant differences in the expression of myosin-heavy chain genes in the LDM of pigs among the four groups (Figure 1a–d). Compared to the CON group, the mRNA expression of *MYH1* of pigs with restricted feeding allowance was significantly decreased, and the mRNA expression of *MYH2* was significantly increased (*p* < 0.05). Under the condition of restricted feeding, the mRNA expression of *MYH1* decreased in a linear and quadratic (*p* < 0.05) manner as dietary energy density increased, and the mRNA expression of *MYH2* linearly increased (*p* < 0.05).

### 3.7. mRNA Expression of Lipid-Metabolism-Related Genes

There were significant differences in the expression of genes related to lipid metabolism in the LDM of experimental pigs among four groups (Figure 2a–d and Figure 3a–d). Compared to the CON group, the mRNA expression of *CD36* and *PNPLA2* in the NE 2330 group was significantly increased (*p* < 0.05), and the mRNA expression of *CPT1B* in both NE 2330 group and NE 2370 group pigs significantly increased (*p* < 0.05). The mRNA expression of *LPL* and *SREBF1* of restricted-fed pigs was significantly higher than those of the CON group pigs (*p* < 0.05). Under the condition of restricted feeding, the mRNA expression of *CD36*, *LPL*, *PNPLA2*, *CPT1B*, and *SREBF1* significantly decreased linearly as dietary energy density increased (*p* < 0.05).

### 3.8. Cecal Microbiota

Compared to the CON group, the Chao1 and Shannon indexes in the NE 2410 group were significantly decreased (*p* < 0.05) (Table 8). Specifically, the Chao1 index decreased in a linear and quadratic manner (*p* < 0.05), while the Shannon index decreased in a quadratic manner as energy density increased (*p* < 0.05).

The microbial community of pigs in the CON group was relatively concentrated. The distribution of the NE 2330 group was relatively scattered and overlapped with the CON group. With the increase in dietary energy density, the microbial community structure of the NE 2370 and NE 2410 groups changed significantly and the differences from other groups gradually appeared (Figure 4).

Cecal microbiota of pigs was dominated by two phyla: Firmicutes and Bacteroidota (Table 9 and Figure 5a). Compared to the CON group pigs, the abundance of Firmicutes in both the NE 2330 and NE 2370 group pigs was significantly decreased (*p* < 0.05), and the abundance of Proteobacteria and Euryarchaeota were significantly increased (*p* < 0.05). As shown in Figure 5c, the abundance of Firmicutes in restricted-fed pigs increased quadratically (*p* < 0.05) and the abundance of Proteobacteria and Euryarchaeota showed a quadratic decrease (*p* < 0.05) as dietary energy density increased.

The dominant genera in cecal chyme were *Streptococcus*, *Lactobacillus*, and *Clostridium_sensu_stricto_1* (Table 10 and Figure 5b). The abundance of *Streptococcus* in treatment groups of pigs was significantly lower than that in the CON group (*p* < 0.05). Compared to the CON group, the abundance of *Escherichia-Shigella* in the NE 2330 group and the abundance of *Methanobrevibacter* and *Treponema* in both the NE 2330 and NE 2370 groups were significantly increased (*p* < 0.05), and the abundance of *Clostridium_sensu_stricto_1* in the NE 2410 group and the abundance of *Christensenellaceae_R-7_group* in the NE 2330 group were significantly decreased (*p* < 0.05). As shown in Figure 5d, the abundance of *Clostridium_sensu_stricto_1* decreased linearly (*p* < 0.05), the abundance of *Streptococcus* and *Christensenellaceae_R-7_group* decreased quadratically (*p* < 0.05), and the abundance of *Escherichia-Shigella*, *Methanobrevibacter*, and *Treponema* increased quadratically (*p* < 0.05) with increased energy density.

### 3.9. Correlation of Between Microbial Diversity, Growth Performance, and Carcass Traits

The chao1 index was negatively correlated with F/G (*p* < 0.05), but positively correlated with dressing percentage and loin muscle area (*p* < 0.05). The observed_features was negatively correlated with F/G (*p* < 0.05). The Shannon index was negatively correlated with F/G (*p* < 0.05), but positively correlated with Initial body weight, dressing percentage, and loin muscle area (*p* < 0.05). The Simpson index was negatively correlated with F/G (*p* < 0.05), but positively correlated with initial body weight, dressing percentage, and loin muscle area (*p* < 0.05) (Figure 6).

## 4. Discussion

The feed to gain ratio of growing-finishing pigs largely depends on the dietary energy allocation to lean or fat tissues [20]. Restricted feeding regimes have been adopted in practical pig production to decelerate fat deposition and promote lean deposition, thereby improving feeding efficiency [21]. In the present study, the final body weight, ADG, and ADFI of restricted-fed pigs were significantly decreased compared to pigs in the CON group, which was consistent with previous findings showing that reduced feeding frequency decreased growth rate but increased the feeding efficiency of pigs [1,22]. Previous studies showed that higher dietary energy concentration could increase ADG and decrease ADFI in pigs [8]. Our study found that increasing dietary NE density improved the ADG and feed efficiency of pigs without affecting ADFI. Additionally, in the current study, restricted-fed pigs exhibited the highest feeding efficiency when the energy density was set at 2370 kcal NE kg^−1^, indicating that the increase in energy density was not simply linearly compensating for the insufficient intake-induced impairment in F/G [23].

Previous findings have demonstrated that modulating feeding frequency could alter the partitioning of energy towards the deposition of lean and fat tissue [4,24]. The present study indicated that restricted feeding significantly reduced dressing percentage, backfat thickness, and loin muscle area, which was consistent with previous evidence suggesting that reduced feeding frequency could shape carcass composition by reducing fat deposition [5,24]. An increase in dietary energy density led to improvements in carcass parameters, indicating that higher energy supply could partially counteract the negative impacts of feeding restriction on carcass traits [25,26]. Color is the most important attribute of meat quality perceived by consumers. In this study, reduced feeding frequency significantly decreased the L* of LDM, which was consistent with previous findings that lower L* in muscle of pigs with longer feeding intervals [27]. It was reported that an increase in dietary energy contributed to an elevation in the b* of chicken meat and pork [28]. Likewise, in the current study, higher dietary energy increased the b* of muscle in restricted-fed pigs. Therefore, controlling the energy density of diets in the range of 2370–2410 kcal NE kg^−1^ under the restricted feeding regime can not only avoid the excessive fat deposition caused by higher energy intake, but also improve the color of the pork, aligning with consumer purchase preferences.

The content of dietary nutrients largely determined the nutritional value of pork [29]. Consistent with previous findings, the results of our study showed that reducing the feeding frequency could decrease the intramuscular fat (IMF) content in pigs [30]. However, IMF levels increased with the increase in energy density, indicating that elevating dietary energy density can restore restricted feeding regime-caused IMF reduction. Fatty acid composition analysis showed that the proportion of major fatty acids was unchanged, indicating the adjustment of energy density had a limited effect on muscle fatty acid profile, even under restricted feeding conditions. The results were consistent with a previous study [31], possibly because the increase in dietary energy density did not reach the threshold for significant changes in fatty acid composition. In the present study, treatment groups exhibited increased expression of lipid metabolism-related genes, including *CD36*, *LPL*, *PNPLA2*, *CPT1B*, and *SREBF1*, which was consistent with the previous finding that increasing the frequency of feeding decreases the expression of fat metabolism genes [32]. The increase in dietary energy density significantly down-regulated the expression of *CD36*, *LPL*, *PNPLA2*, *CPT1B*, and *SREBF1* in restricted-fed pigs, which was similar to previous evidence demonstrating that higher dietary energy density upregulated the expression of these lipogenic genes and facilitated lipid metabolism [33].

In this study, Chao1 and Shannon indices were significantly decreased in the NE 2410 group compared to the CON group, which was consistent with the previous findings that intestinal flora alpha diversity was reduced in mice fed a high-fat and high-sugar diet [34]. The reduction in microbial diversity directly impairs nutrient metabolism efficiency. High diversity means that the microbiota can more comprehensively break down complex carbohydrates, lipids, and other nutrients in the feed. The excessively high energy concentration in the NE 2410 group may have inhibited the abundance of beneficial degradative bacteria, leading to a decreased ability of the microbiota to break down dietary fiber and synthesize SCFAs. The bacteria falling in phylum Firmicutes has been associated with enhanced fat deposition in animals [35]. In this study, the abundance of Firmicutes in pigs of the NE 2330 group was decreased compared to the CON group. However, under restricted feeding conditions, the abundance of Firmicutes increased continuously with the elevation of dietary energy density, indicating that the increase in energy density led to an increase in backfat thickness in restricted-fed pigs by increasing the abundance of Firmicutes. Phyla Proteobacteria and Euryarchaeota have been associated with reduced fat deposition in hosts, as indicated by higher abundances of these two phyla in rodents with lower body fat percentage [36,37]. In this study, the abundances of Proteobacteria and Euryarchaeota in pigs of the NE 2330 group were increased compared to the CON group. Additionally, the abundances of Proteobacteria and Euryarchaeota in restricted-fed pigs were decreased as dietary energy density increased. Genus *Christensenellaceae_R-7_group* has been recognized as a biomarker improving the backfat thickness of pigs [38]. In this study, a restricted feeding regime significantly reduced the *Christensenellaceae_R-7_group* abundance, while the elevation of dietary energy density increased the *Christensenellaceae_R-7_group* abundance, indicating that the changes in backfat thickness of pigs in response to feeding regime and dietary energy density might be attributed to the altered proportion of this taxon. Consistent with previous findings, our study found that higher dietary energy increased the abundance of *Methanobrevibacter* in restricted-fed pigs [39]. Previous studies showed that the abundances of *Treponema* were negatively associated with lipid deposition in pigs. Our study found that pigs in the NE 2410 group had a lower level of genus *Treponema* in cecal content than pigs in the NE 2330 and NE 2370 groups, indicating that the increase in dietary energy density might lead to an increase in backfat thickness in restricted-fed pigs that was associated with decreased abundance of *Treponema*.

## 5. Conclusions

Reduced feeding frequency significantly decreased growth rate, carcass weight, and IMF while improving F/G and reducing LDM lightness. As dietary NE density increased from 2330 to 2410 kcal kg^−1^, the fat deposition and carcass traits were improved. These changes were associated with the alterations in intestinal microbial composition and lipid-metabolism-related gene expression. Taken together, under restricted feeding conditions, a dietary NE level of 2370 kcal kg^−1^ enhanced feed efficiency, while improving carcass traits and IMF deposition of LDM in growing-finishing pigs.

## Figures and Tables

**Figure 1 animals-15-03514-f001:**
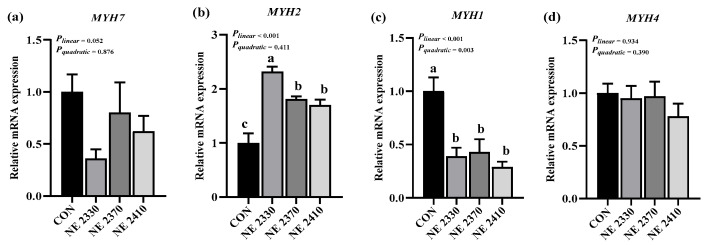
Effects of dietary energy density on mRNA expression of myosin heavy chain genes (**a**–**d**). (**a**) *MYH7*, myosin heavy chain 7; (**b**) *MYH2*, myosin heavy chain 2; (**c**) *MYH1*, myosin heavy chain 1; (**d**) *MYH4*, myosin heavy chain 4. ^a,b,c^ Different letters on the bars indicate significant differences between groups.

**Figure 2 animals-15-03514-f002:**
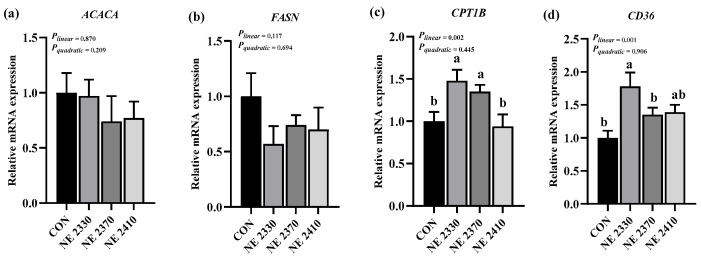
Effects of dietary energy density on mRNA expression of lipid-metabolism-related genes (**a**–**d**). (**a**) *ACACA*, acetyl-CoA carboxylase alpha; (**b**) *FASN*, fatty acid synthase; (**c**) *CPT1B*, carnitine palmitoyltransferase 1B; (**d**) *CD36*, CD36 molecule. ^a,b^ Different letters on the bars indicate significant differences between groups.

**Figure 3 animals-15-03514-f003:**
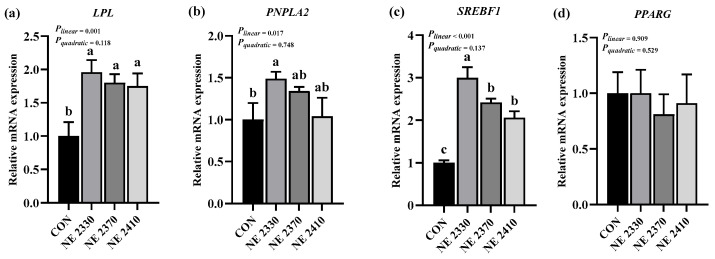
Effects of dietary energy density on mRNA expression of lipid-metabolism-related genes (**a**–**d**). (**a**) *LPL*, lipoprotein lipase; (**b**) *PNPLA2*, patatin-like phospholipase domain containing 2; (**c**) *SREBF1*, sterol regulatory element binding transcription factor 1; (**d**) *PPARG*, peroxisome proliferator-activated receptor gamma. ^a,b,c^ Different letters on the bars indicate significant differences between groups.

**Figure 4 animals-15-03514-f004:**
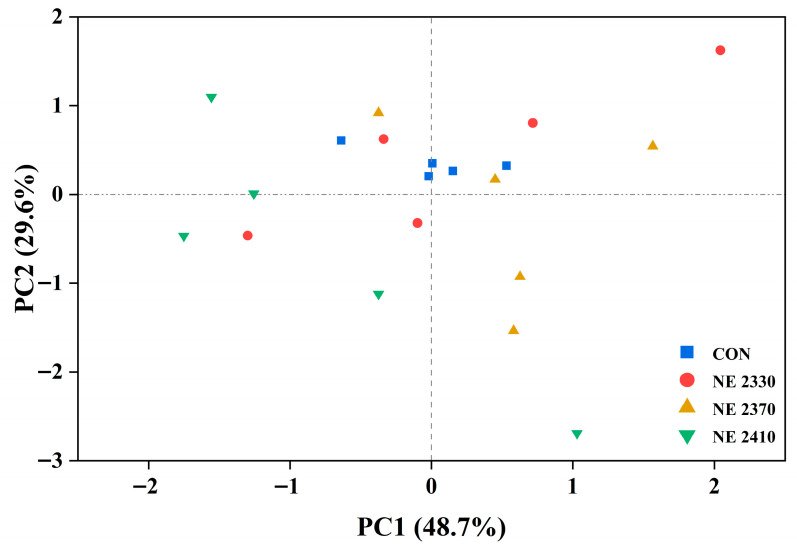
Principal component analysis (PCA) plot based on ASV composition.

**Figure 5 animals-15-03514-f005:**
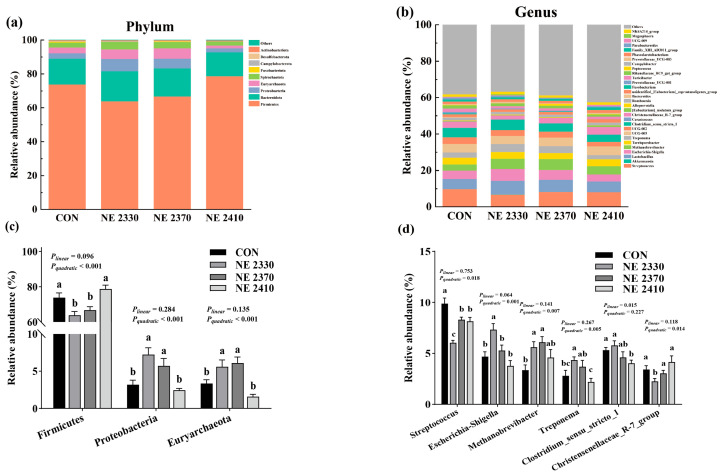
Effects of dietary energy density on cecal microbiota in pigs with restricted feed allowance. (**a**) Relative abundance of cecal microbial communities at the phylum level in pigs. (**b**) Relative abundance of cecal microbial communities at the genus level in pigs. (**c**) The differential bacteria of microbes at the phylum level. (**d**) The differential bacteria of microbes at the genus level. ^a,b,c^ Different letters on the bars indicates significant differences between groups.

**Figure 6 animals-15-03514-f006:**
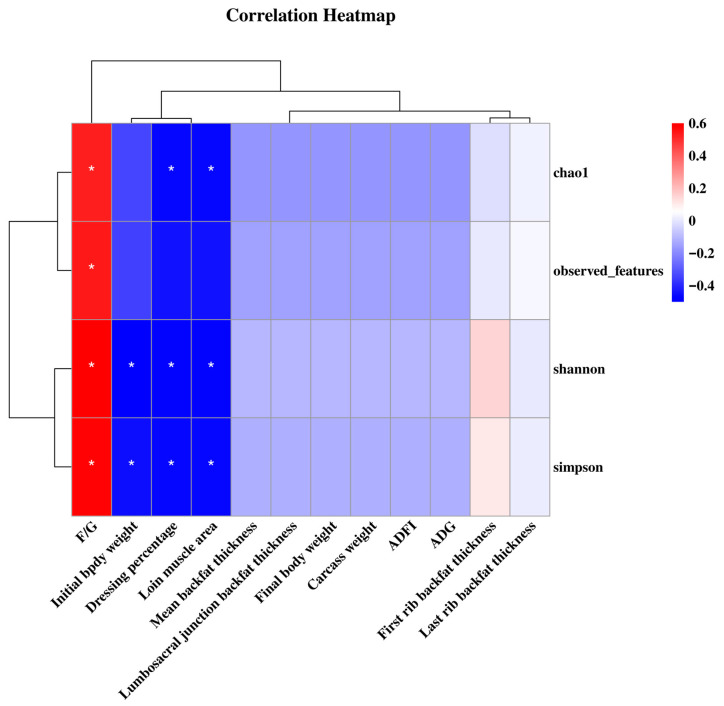
Correlation heatmap between microbiota and growth performance and carcass traits. Spearman correlations were applied. ADG, average daily gain; ADFI, average daily feed intake; F/G, feed to gain ratio. * *p* < 0.05.

**Table 1 animals-15-03514-t001:** Ingredient composition and nutrient levels of experimental diets (%, as-fed basis).

Items	Diet (Kcal NE kg^−1^)		
	NE 2330	NE 2370	NE 2410
Corn	42.16	48.79	41.05
Soybean meal (43%)	8.61	11.71	8.73
Wheat flour	15.00	15.00	15.00
Fermented soybean meal	15.00	15.00	15.00
Corn germ meal	10.00	3.95	10.00
Soybean hull	0.00	2.00	0.00
Wheat bran	5.78	0.00	5.00
Soybean oil	0.00	0.00	1.79
Limestone	1.03	0.94	0.81
lysine-HCl (70%)	0.72	0.63	0.71
Monocalcium phosphate	0.52	0.65	0.55
Salt	0.30	0.30	0.30
Vitamin-mineral premix ^1^	0.60	0.77	0.78
L-threonine	0.19	0.17	0.18
DL-methionine	0.09	0.09	0.10
Nutrient level ^2^			
Net energy (Kcal kg^−1^)	2330	2370	2410
Crude protein (%)	14.5	14.5	14.5
Ether extract (%)	2.63	2.59	4.34
Crude ash (%)	3.92	3.84	4.14

^1^ The premix provided the following per kg of diets 0 mg: V_A_ 5512 IU, V_D3_ 2200 IU, V_E_ 30 IU, V_K3_ 2.2 mg, V_B1_ 1.5 mg, V_B2_ 3.0 mg, V_B6_ 3.0 mg, V_B12_ 27.6 μg, pantothenic acid 14.0 mg, nicotinic acid 30.0 mg, choline 400.0 mg, folic acid 0.7 mg, biotin 44.0 μg, Mn 40.0 mg, Fe 75.0 mg, Zn 75.0 mg, Cu 20.0 mg, I 0.3 mg, Se 0 mg. ^2^ Nutrient levels were all calculated values.

**Table 2 animals-15-03514-t002:** RT-qPCR gene primer sequence.

Gene ^1^	Accession Number	Primer Sequences (5′–3′)	Annealing Temperature (°C)	Size/bp
*ACTB*	XM_021086047.1	F: TCTGGCACCACACCTTCT	56.33	114
		R: TGATCTGGGTCATCTTCTCAC		
*MYH7*	NM_213855.2	F: AAGACCCGCTCAACGAGACAGTGG	57.55	121
		R: GCCTTGCCTTTGCCCTTCTCAACA		
*MYH2*	NM_214136.1	F: TCATCAGTGCCAACCCGCTG	55.22	120
		R: AAGCCAGTTTTCCTGTAGTGCCAAA		
*MYH4*	NM_001123141.1	F: TTGAGGAGTTAAAGAGGCAGCTAGAAGAGG	56.81	166
		R: TCGCTGTTGGCCTTGGACATTGC		
*MYH1*	NM_001104951.2	F: TGGAGGCCAGGGTACGTGAA	55.74	134
		R: CTTGCGGTCTTCCTCAGTTTGGT		
*ACACA*	XM_054315912.1	F: TGTCCACTCAAGCATACCTCCCA	56.17	136
		R: GCTACCATGCCAATCTCATTTCCTCC		
*FASN*	XM_054315477.1	F: GCCGAGTACAGCGTCAACAACC	57.26	173
		R: TGGTCCTTCTTCATCAGCGGGAT		
*CPT1B*	NM_0002478.1	F: AGTCATGGTGGGCGACTAACTATGTG	55.42	168
		R: ATCATGGCGTGGACAGCGTTC		
*LPL*	NM_000237.3	F: AACGTCATTGTGGTGGACTGGCT	56.50	165
		R: TCCAAGGCTGTATCCCAGGAGGTG		
*CD36*	XM_021102279.1	F: CTGTGGACTCATTGCTGGTGCTG	55.93	179
		R: AAAACTGTCTGTAAACTTCCGTGCCT		
*PNPLA2*	NM_001098605.1	F: CCTGCCTCTCTACGAACTCAAGAGC	56.45	132
		R: AGGCTGAACTGGATGCTGGTGT		
*SREBF1*	NM_214157.1	F: CGCAAGACGGCGGATTTA	58.32	218
		R: GCGACGGTGCCTCTGGTAGT		
*PPARG*	XM_005669788.3	F: TCCAGCATTTCCACTCCACAC	57.18	127
		R: GGGACACAGGCTCCACTTTG		

^1^ *ACTB*, actin beta; *MYH7*, myosin-heavy chain 7; *MYH4*, myosin-heavy chain 4; *MYH2*, myosin-heavy chain 2; *MYH1*, myosin-heavy chain 1; *ACACA*, acetyl-CoA carboxylase alpha; *FASN*, fatty acid synthase *CPT1B*, carnitine palmitoyltransferase 1B; *LPL*, lipoprotein lipase; *PNPLA2*, patatin-like phospholipase domain containing 2; *SREBF1*, sterol regulatory element binding transcription factor 1; *PPARG*, peroxisome proliferator-activated receptor gamma.

**Table 3 animals-15-03514-t003:** Effects of dietary energy density on growth performance of pigs with restricted feed allowance.

Items ^1^	CON ^2^	Dietary Energy Density for Non-AD Pigs ^3^			SEM	*p*-Value		
		NE 2330	NE 2370	NE 2410		ANOVA	Linear	Quadratic
Initial body weight, kg	25.99	26.09	25.99	25.87	0.56	0.997	0.903	0.868
Final body weight, kg	121.22 ^a^	111.12 ^b^	116.80 ^a^	118.16 ^a^	1.59	0.007	<0.001	0.657
ADG, g/d	721.45 ^a^	644.19 ^c^	687.97 ^b^	699.17 ^ab^	8.64	<0.001	<0.001	0.595
ADFI, g/d	2576.15 ^a^	2170.98 ^b^	2201.19 ^b^	2191.51 ^b^	17.76	<0.001	<0.001	<0.001
F/G	3.57 ^a^	3.37 ^b^	3.20 ^c^	3.13 ^c^	0.03	<0.001	<0.001	<0.001

Results are presented as mean and SEM (*n* = 8). ^a,b,c^ Within a row with different superscripts differ significantly (*p* < 0.05). ^1^ ADG, Average daily gain; ADFI, Average daily feed intake; F/G, Feed to gain ratio. ^2^ CON, basal diet. ^3^ AD, ad libitum feeding.

**Table 4 animals-15-03514-t004:** Effects of dietary energy density on carcass traits of pigs with restricted feed allowance.

Items	CON ^1^	Dietary Energy Density for Non-AD Pigs ^2^			SEM	*p*-Value		
		NE 2330	NE 2370	NE 2410		ANOVA	Linear	Quadratic
Carcass weight, g	91.79 ^a^	82.78 ^c^	87.43 ^b^	88.51 ^ab^	1.23	<0.001	<0.001	0.582
Dressing percentage, %	75.72 ^a^	74.48 ^b^	74.85 ^b^	74.91 ^b^	0.15	<0.001	<0.001	0.149
First rib backfat thickness, mm	5.55 ^a^	4.31 ^c^	4.82 ^b^	4.95 ^b^	0.15	<0.001	<0.001	0.775
Last rib backfat thickness, mm	3.40 ^a^	2.44 ^b^	2.77 ^b^	2.92 ^ab^	0.21	0.028	0.004	0.728
Lumbosacral junction backfat thickness, mm	3.60 ^a^	2.44 ^b^	3.01 ^b^	3.11 ^b^	0.18	<0.001	<0.001	0.804
Mean backfat thickness, mm	4.18 ^a^	3.06 ^c^	3.53 ^b^	3.66 ^b^	0.12	<0.001	<0.001	0.836
Loin muscle area, cm^2^	35.04 ^a^	32.64 ^b^	34.17 ^ab^	35.31 ^a^	0.72	0.060	0.015	0.225

Results are presented as mean and SEM (*n* = 8). ^a,b,c^ Within a row with different superscripts differ significantly (*p* < 0.05). ^1^ CON, basal diet. ^2^ AD, ad libitum feeding.

**Table 5 animals-15-03514-t005:** Effects of dietary energy density on meat quality of pigs with restricted feed allowance.

Items ^1^	CON ^2^	Dietary Energy Density for Non-AD Pigs ^3^			SEM	*p*-Value		
		NE 2330	NE 2370	NE 2410		ANOVA	Linear	Quadratic
pH_45min_	5.67	6.13	5.96	5.88	0.12	0.081	0.386	0.033
L*_45min_	39.13 ^a^	35.78 ^b^	37.56 ^ab^	36.94 ^b^	0.73	0.024	0.152	0.071
a*_45min_	5.79	4.86	5.78	4.72	0.39	0.117	0.201	0.864
b*_45min_	4.23 ^a^	3.33 ^b^	3.77 ^ab^	3.80 ^ab^	0.19	0.025	0.330	0.023
Drip loss, %	5.38	6.15	4.65	4.63	0.61	0.259	0.179	0.521
Cooking loss, %	35.60	38.67	35.21	37.93	1.14	0.105	0.494	0.882

Results are presented as mean and SEM (*n* = 8). ^a,b^ Within a row with different superscripts differ significantly (*p* < 0.05). ^1^ L*, lightness; a*, redness; b*, yellowness. ^2^ CON, basal diet. ^3^ AD, ad libitum feeding.

**Table 6 animals-15-03514-t006:** Effects of dietary energy density on chemical composition analysis in the LDM of pigs with restricted feed allowance.

Items ^1^	CON ^2^	Dietary Energy Density for Non-AD Pigs ^3^			SEM	*p*-Value		
		NE 2330	NE 2370	NE 2410		ANOVA	Linear	Quadratic
Moisture (%)	70.01	70.55	70.90	71.16	0.29	0.398	0.543	0.141
CP (%)	24.41	23.96	22.36	23.35	0.81	0.326	0.197	0.383
EE (%)	7.91 ^a^	5.42 ^b^	7.91 ^a^	8.38 ^a^	0.75	0.039	0.258	0.059

Results are presented as mean and SEM (*n* = 8). ^a,b^ Within a row with different superscripts differ significantly (*p* < 0.05). ^1^ CP, crude protein; EE, ether extract. ^2^ CON, basal diet. ^3^ AD, ad libitum feeding.

**Table 7 animals-15-03514-t007:** Effects of dietary energy density on fatty acid profile in the LDM of pigs with restricted feed allowance.

Items	CON ^1^	Dietary Energy Density for Non-AD Pigs ^2^			SEM	*p*-Value		
		NE 2330	NE 2370	NE 2410		ANOVA	Linear	Quadratic
C8:0, %	0.51	0.79	0.92	1.00	0.09	0.209	0.319	0.075
C10:0, %	0.07	0.06	0.06	0.08	0.01	0.142	0.128	0.25
C12:0, %	0.08	0.09	0.07	0.09	0.01	0.277	0.927	0.808
C14:0, %	1.32	1.32	1.35	1.36	0.02	0.842	0.958	0.379
C16:0, %	27.66	27.66	27.51	27.66	0.18	0.990	0.936	0.845
C16:1, %	2.91	3.07	3.05	3.26	0.07	0.416	0.669	0.278
C17:0, %	0.1	0.12	0.11	0.11	0.01	0.496	0.203	0.476
C18:0, %	13.73	13.52	13.85	13.63	0.16	0.901	0.775	0.722
C18: ln9c, %	45.69	43.84	44.06	43.69	0.37	0.209	0.117	0.224
C18:2n6c, %	5.86	6.9	6.58	6.4	0.29	0.656	0.219	0.854
C18:3n-3, %	0.19	0.22	0.18	0.2	0.008	0.467	0.418	0.322
C20:0, %	0.15	0.15	0.16	0.16	0.01	0.897	0.885	0.456
C20:1, %	0.65	0.7	0.7	0.7	0.02	0.738	0.386	0.531
C20:2, %	0.19	0.22	0.22	0.21	0.01	0.447	0.142	0.454
C20:3n-6, %	0.12	0.16	0.15	0.16	0.01	0.424	0.215	0.376
C20:4n-6, %	0.78	1.05	1.02	1.11	0.08	0.493	0.325	0.347
C22:1n-9, %	0.08	0.09	0.08	0.1	0.01	0.501	0.784	0.832
Total SFAs ^3^, %	43.47	43.55	43.88	43.96	0.27	0.905	0.944	0.470
Total UFAs ^4^, %	56.39	56.17	55.96	55.75	0.26	0.857	0.861	0.444
Total MUFAs, %	49.33	47.71	47.88	47.75	0.36	0.356	0.153	0.337
Total PUFAs, %	7.14	8.55	8.15	8.08	0.37	0.623	0.220	0.722
n-3 PUFA, %	0.19	0.22	0.18	0.2	0.01	0.467	0.417	0.322
n-6 PUFA, %	6.76	8.11	7.75	7.68	0.36	0.632	0.226	0.712
SFA/MUFA, %	0.88	0.91	0.92	0.92	0.01	0.541	0.341	0.293
SFA/PUFA, %	6.26	5.32	6.03	5.74	0.27	0.664	0.319	0.870
UFA/SFA, %	1.3	1.29	1.28	1.27	0.01	0.899	0.886	0.483
n6/n3, %	35.4	38.61	42.41	39.75	1.76	0.613	0.453	0.268

Results are presented as mean and SEM (*n* = 8). ^1^ CON, basal diet. ^2^ AD, ad libitum feeding. ^3^ Total SFAs = C8:0 + C10:0 + C12:0 + C14:0 + C16:0 + C17:0 + C18:0 + C20:0 [19]. ^4^ Total UFAs = C16:1 + C18:1n9c + C18:2n-6c + C18:3n-3 + C20:1 + C20:2 + C20:3n-6 +C20:4n-6 + C22:1n-9 [19].

**Table 8 animals-15-03514-t008:** Effects of dietary energy density on alpha diversity of cecal microbes in pigs with restricted feed allowance.

Items	CON ^1^	Dietary Energy Density for Non-AD Pigs ^2^			SEM	*p*-Value		
		NE 2330	NE 2370	NE 2410		ANOVA	Linear	Quadratic
Chao1	737.81 ^ab^	811.31 ^a^	688.56 ^b^	580.23 ^c^	33.59	<0.001	<0.001	<0.001
Observed_features	726.50	736.40	655.60	566.60	46.89	0.073	0.579	0.021
Shannon	7.27 ^ab^	7.38 ^a^	6.61 ^b^	5.35 ^c^	0.23	<0.001	0.070	<0.001
Simpson	0.98	0.98	0.97	0.90	0.01	0.062	0.201	0.002

Results are presented as mean and SEM (*n* = 5). ^a,b,c^ Within a row with different superscripts differ significantly (*p* < 0.05). ^1^ CON, basal diet. ^2^ AD, ad libitum feeding.

**Table 9 animals-15-03514-t009:** Effect of different energy density on the composition of dominant fecal microbiota at the phylum level under restricted feeding conditions (relative abundance > 1%).

Group	Phylum	Relative Abundance (%)
CON **^1^**	Firmicutes	73.82
	Bacteroidota	15.21
	Euryarchaeota	3.34
	Proteobacteria	3.18
	Spirochaetota	2.89
NE 2330	Firmicutes	63.80
	Bacteroidota	17.76
	Proteobacteria	7.23
	Euryarchaeota	5.60
	Spirochaetota	4.42
NE 2370	Firmicutes	66.75
	Bacteroidota	16.56
	Euryarchaeota	6.10
	Proteobacteria	5.71
	Spirochaetota	3.72
NE 2410	Firmicutes	78.72
	Bacteroidota	14.03
	Proteobacteria	2.44
	Spirochaetota	2.36
	Euryarchaeota	1.58

^1^ CON, basal diet.

**Table 10 animals-15-03514-t010:** Effect of different energy density on the abundances of dominant genera under restricted feeding conditions (relative abundance > 2%).

Group	Genus	Relative Abundance (%)
CON ^1^	*Streptococcus*	9.88
	*Lactobacillus*	5.43
	*Clostridium_sensu_stricto_1*	5.31
	*UCG-005*	4.69
	*Escherichia-Shigella*	4.66
	*Terrisporobacter*	3.78
	*UCG-002*	3.57
	*Christensenellaceae_R-7_group*	3.41
	*Methanobrevibacter*	3.34
	*Treponema*	2.79
NE 2330	*Lactobacillus*	7.43
	*Streptococcus*	7.23
	*Escherichia-Shigella*	6.52
	*Clostridium_sensu_stricto_1*	5.77
	*Methanobrevibacter*	5.60
	*UCG-005*	4.35
	*Treponema*	3.94
	*Terrisporobacter*	3.75
	*UCG-002*	3.25
	*Christensenellaceae_R-7_group*	2.25
NE 2370	*Streptococcus*	8.30
	*Lactobacillus*	6.64
	*Methanobrevibacter*	6.10
	*Escherichia-Shigella*	5.27
	*UCG-005*	4.73
	*Clostridium_sensu_stricto_1*	4.60
	*Treponema*	3.69
	*Terrisporobacter*	3.28
	*UCG-002*	3.24
	*Christensenellaceae_R-7_group*	3.04
NE 2410	*Streptococcus*	8.14
	*Lactobacillus*	5.94
	*UCG-005*	4.79
	*Methanobrevibacter*	4.58
	*Christensenellaceae_R-7_group*	4.14
	*Clostridium_sensu_stricto_1*	4.02
	*Terrisporobacter*	3.84
	*Escherichia-Shigella*	3.77
	*UCG-002*	2.40
	*Treponema*	2.20

^1^ CON, basal diet.

## Data Availability

The authors confirm that the data supporting the findings of this study are available within the article. The raw sequencing data was deposited in the NCBI BioProject database with the accession number PRJNA1275840.

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
