# Peer review of "Dietary Net Energy Concentration Affects Growth Performance, Carcass Traits, Intramuscular Fatty Acid Profile, and Cecal Microbiota of Pigs with Restricted Feed Allowance"

_animals, 2025, doi:10.3390/ani15243514_

Round 1

Reviewer 1 Report

Comments and Suggestions for Authors

The study is interesting and provides valuable insights for the swine industry. I commend the authors for their clear and concise presentation. The findings are particularly relevant for producers seeking to enhance productivity and profitability.

Below are my minor comments for revision:

  1. Title: Replace “microbiome” with “microbiota.”

  2. Line 36: Change “fed-restricted” to “feed restricted.”

  3. Line 37: Please specify the breed of pigs used.

  4. Line 71: Include a reference for the term “absolute growth rate.”

  5. Figures 1 and 2: Improve the resolution for better clarity.

  6. Line 356: Write out “F/G” in full.

Author Response

Comments 1: Title: Replace “microbiome” with “microbiota.”

Response 1: Thank for your suggestion. We have replaced “microbiome” with “microbiota.”See line 3.

Comments 2: Line 36: Change “fed-restricted” to “feed restricted.”

Response 2: Thank for your suggestion. We have changed “fed-restricted” to “feed restricted.”See line 36.

Comments 3: Line 37: Please specify the breed of pigs used.

Response 3: Thank you for your suggestion. We have added the breed of the pig. See line 37.

Comments 4: Line 71: Include a reference for the term “absolute growth rate.”

Response 4: Thank you for your reminder regarding the definition of “absolute growth rate.” We have changed “absolute growth rate” to “growth rate” to make it easier for readers to understand. See line 70.

Comments 5: Figures 1 and 2: Improve the resolution for better clarity.

Response 5: Thank you for your valuable suggestions. Figures 1 and 2 have been adjusted accordingly. See Figures 1, 2, 3 and 4.

Comments 6: Line 356: Write out “F/G” in full.

Response 6: Thank you for pointing out this oversight. We agree with your comment and have used the full term 'F/G' in the revised manuscript. See line 374.

Reviewer 2 Report

Comments and Suggestions for Authors

The article fits well within the journal’s scope, presents original content, and has the potential to contribute to the advancement of knowledge in swine production. The methodology and results are significant and were described satisfactorily. Overall, the manuscript is well written, requiring only minor suggestions. Its main weakness lies in the discussion section. Once this issue is addressed, I believe both reader interest and the article’s merit will increase substantially. The English is adequate and easy to understand.

Below are some comments on each section of the manuscript:

Abstract

The abstract is appropriate and follows the guidelines of Animals. Some keywords, such as “growth performance” and “carcass traits”, should be replaced since they already appear in the title.

Introduction

The introduction is concise and effectively presents the research problem. The paragraphs are well structured. However, adding a few sentences to clearly state the study’s hypothesis would strengthen this section and align it with Animals guidelines.

Materials and Methods

The materials and methods are appropriate and reproducible. The techniques, as well as the kits and equipment used, are described in sufficient detail.
It is recommended to avoid phrases such as “The composition and nutrient levels of the experimental diets are shown in Table 1” and instead integrate them more naturally into the text. For example: “All diets were iso-nitrogenous, with identical crude protein content (Table 1).” Table 1 is used as an example, but this applies to most tables.

Results

The results are clearly described and presented in a logical and coherent manner. All tables are self-explanatory and properly cited in the text.
Figure 1 could be divided into more than one figure, as it is too small and becomes distorted when zoomed in. This could be addressed in later stages, but the suggestion is worth considering.

Discussion

The discussion could be improved. Many variables were analyzed only by comparing the current results with those reported in the literature. Including a stronger biological context explaining why certain results occurred would enrich the discussion. For instance: how do changes in the microbiome affect specific metabolic pathways in the animal?

Conclusion

The conclusion is appropriate and adequately addresses the stated objectives and results.

References

The cited references are relevant; however, most were published more than five years ago (approximately 70%). It would be advisable to update those that are not essential for discussing the results.

Author Response

Comments 1: Abstract

The abstract is appropriate and follows the guidelines of Animals. Some keywords, such as “growth performance” and “carcass traits”, should be replaced since they already appear in the title.

Response 1: Thank you for your suggestion. We have removed these two keywords and updated a new keywords list. See line 51.

Comments 2: Introduction

The introduction is concise and effectively presents the research problem. The paragraphs are well structured. However, adding a few sentences to clearly state the study‘s hypothesis would strengthen this section and align it with Animals guidelines.

Response 2: Thank you for your valuable suggestion. we have added the research hypothesis in the final paragraph of the introduction. See lines 82-83.

Comments 3: Materials and Methods

The materials and methods are appropriate and reproducible. The techniques, as well as the kits and equipment used, are described in sufficient detail.

It is recommended to avoid phrases such as “The composition and nutrient levels of the experimental diets are shown in Table 1” and instead integrate them more naturally into the text. For example: “All diets were iso-nitrogenous, with identical crude protein content (Table 1).” Table 1 is used as an example, but this applies to most tables.

Response 3: Thank you for your suggestion. We modified all table reference sentences to make them integrate more naturally with the text. See lines 92-93 and line 177.

Comments 4: Results

The results are clearly described and presented in a logical and coherent manner. All tables are self-explanatory and properly cited in the text.

Figure 1 could be divided into more than one figure, as it is too small and becomes distorted when zoomed in. This could be addressed in later stages, but the suggestion is worth considering.

Response 4: Thank you for your suggestion. We greatly appreciate your recognition of these aspects. We have made the corresponding adjustments and split figure 1 into three smaller figures. See figures 1, 2, and 3.

Comments 5: Discussion

The discussion could be improved. Many variables were analyzed only by comparing the current results with those reported in the literature. Including a stronger biological context explaining why certain results occurred would enrich the discussion. For instance: how do changes in the microbiome affect specific metabolic pathways in the animal?

Response 5: Thank you for your suggestion. We have already made changes to this part.

Comments 6: The conclusion is appropriate and adequately addresses the stated objectives and results.

Response 6: Thank you for your positive evaluation of the conclusion section. We greatly appreciate your recognition that the conclusion effectively aligns with the study’s stated objectives and summarizes the key results.

Comments 7: References

The cited references are relevant; however, most were published more than five years ago (approximately 70%). It would be advisable to update those that are not essential for discussing the results.

Response 7: Thank you for your suggestion. We have reviewed the references and made adjustments.

Reviewer 3 Report

Comments and Suggestions for Authors

This manuscript presented a well-designed study investigating the effects of dietary net energy concentration on growth performance, carcass traits, intramuscular fatty acids profile and cecal microbiome of pigs with restricted feed allowance. The authors concluded that under twice-daily feeding, a dietary NE level of 2370 kcal kg-1 achieves the best balance between feed efficiency and carcass traits in growing-finishing pigs. Overall, it is a well-written article that, but several critical points should be addressed before publication.

Major issues:

  1. Only castrated male pigs were used as experimental animals, and the impact of gender on energy metabolism, fat deposition, and microbial composition was not considered. It is recommended that the authors clarify the rationale for exclusively using castrated male pigs or add the discussion with gender-related analyses to define the applicable scope of the findings.
  2. In the cecal microbial analysis, alpha diversity (Chao1 and Shannon indices) was significantly decreased in the NE 2410 group; however, the actual impacts of this reduced diversity on pig production performance have not been clearly elaborated. It is recommended that correlation analyses between microbial diversity, growth performance, and carcass traits be supplemented to strengthen the conclusion that changes in microbial composition mediate the effect of net energy concentration.

Minor issues

  1. Line 99, ‘‘the ratio of feed intake to body gain (F/G)’’ was defined mistakenly, please define F/G correctly.
  2. Line 103-106, in all vitamin abbreviations should use subscripted number.
  3. Line 137, change “hung with a metal hook to suspend” into “hung on a metal hook to be suspended”.
  4. Table 1, the “Corn” appeared twice in the same table, and the proportions are exactly the same. Check the experimental ingredient list.
  5. Table 2, add annealing temperature for all the primers.
  6. Table 3, present the definition of abbreviation of F/G rather than the calculation equation. And all the tables line color should adjust into black.
  7. Line 186, the concentration of agarose gel?
  8. Lines 282-284, change ‘‘the MYH2 mRNA expression’’ into “the mRNA expression of MYH2’’.
  9. Lines 197-198, Lines 294-330, rewrite the sentences.
  10. The data were not presented as mean and SEM in the figures, modify the figure captions. Change “Within a row with different superscripts differ significantly” into “Different letters on the bars indicates significant differences between groups”.
  11. Line 272-273, provide the references for calculating total SFAs and UFAs.
  12. Figure caption of Figure 3 (c) and (d), capitalized the first letter of the initial word.
  13. Line 355, don’t use abbreviation to initiate a sentence.
  14. Conclusion section, meat quality should be considered in making a conclusion.
  15. Several references appeared to have incomplete information. It is recommended that the authors carefully verify and supplement all required bibliographic details to ensure compliance with the journal’s reference formatting guidelines.

Author Response

Comments 1: Only castrated male pigs were used as experimental animals, and the impact of gender on energy metabolism, fat deposition, and microbial composition was not considered. It is recommended that the authors clarify the rationale for exclusively using castrated male pigs or add the discussion with gender-related analyses to define the applicable scope of the findings.

Response 1: Thank you for the reviewers' attention to the experimental design. Regarding the reasons for selecting only castrated boars and the applicability of the results, we provide the following explanations:

This study focuses on the commercial breeding scenario of Sichuan-Tibetan black pigs. In actual production, castrated boars are the dominant breeding type due to their stable growth performance and superior meat quality. The selection of castrated boars ensures that the research results are more aligned with practical production needs and provides direct guidance for breeding practices.

There are significant differences in energy metabolism and fat deposition between sows and intact boars (e.g., sows have higher IMF content than intact boars, while intact boars exhibit faster growth rates than sows). However, castrated boars have relatively stable metabolic characteristics and are less affected by fluctuations in sex-related hormones. Using castrated boars minimizes interference from sex hormones, thereby improving the accuracy and reliability of the experimental results.

In summary, the selection of castrated boars is based on considerations of practical production applicability and experimental accuracy. The results are relevant to commercial breeding scenarios, and we have clarified directions for future research to expand the universality of the findings.

Comments 2: In the cecal microbial analysis, alpha diversity (Chao1 and Shannon indices) was significantly decreased in the NE 2410 group; however, the actual impacts of this reduced diversity on pig production performance have not been clearly elaborated. It is recommended that correlation analyses between microbial diversity, growth performance, and carcass traits be supplemented to strengthen the conclusion that changes in microbial composition mediate the effect of net energy concentration.

Response 2: Thank you for your insightful comments and suggestions. We have added an analysis of the correlation between microbiota and growth performance as well as carcass traits.

Comments 3: Line 99, “the ratio of feed intake to body gain (F/G)” was defined mistakenly, please define F/G correctly.

Response 3: Thank you for your suggestion. We have made the corresponding revisions. See line 99.

Comments 4: Line 103-106, in all vitamin abbreviations should use subscripted number.

Response 4: Thank for your suggestion. We have made the revisions. See lines 104 and 105.

Comments 5: Line 137, change “hung with a metal hook to suspend” into “hung on a metal hook to be suspended”.

Response 5: Thank for your suggestion. We have changed “hung with a metal hook to suspend” into “hung on a metal hook to be suspended”. See line 137.

Comments 6:Table 1, the “Corn” appeared twice in the same table, and the proportions are exactly the same. Check the experimental ingredient list.

Response 6: Thank you for your suggestion. We have made the revisions. See table 1.

Comments 7: Table 2, add annealing temperature for all the primers.

Response 7: Thank for your suggestion. We have added the annealing temperature to Table 2. See table 2.

Comments 8: Table 3, present the definition of abbreviation of F/G rather than the calculation equation. And all the tables line color should adjust into black.

Response 8: Thank for your suggestion. We have replaced the calculation equation of “F/G” in Table 3 with its full name and abbreviation definition. See line 221. We have adjusted the line colors of all tables in the manuscript to black.

Comments 9: Line 186, the concentration of agarose gel?

Response 9: Thank you for your attention to the experimental detail regarding the agarose gel concentration. We apologize for the oversight in the original manuscript. We used a 1.5% agarose gel (w/v). See line 186.

Comments 10: Lines 282-284, change “the MYH2 mRNA expression” into “the mRNA expression of MYH2’’.

Response 10: Thank for your suggestion. We have changed “the MYH2 mRNA expression” into “the mRNA expression of MYH2’’. See lines 282-283.

Comments 11: Lines 197-198, Lines 294-330, rewrite the sentences.

Response 11: Thank you for your suggestions on optimizing the expression. To ensure that the revisions improve clarity, fluency, and academic rigor, we have made the necessary changes. See lines 197-198 and Lines 294-330.

Comments 12: The data were not presented as mean and SEM in the figures, modify the figure captions. Change “Within a row with different superscripts differ significantly” into “Different letters on the bars indicates significant differences between groups”.

Response 12: Thank you for your careful review and guidance on the chart presentation. We agree with your suggestions and have addressed both issues. See lines 347 and lines 354-355.

Comments 13: Line 272-273, provide the references for calculating total SFAs and UFAs.

Response 13: Thank for your suggestion. We have added the relevant references. See lines 274-275.

Comments 14: Figure caption of Figure 3 (c) and (d), capitalized the first letter of the initial word.

Response 14: Thank for your suggestion. We agree with your suggestion and have promptly revised the captions for Figure 3 (c) and (d) in the revised manuscript, ensuring the first letter of the initial word is capitalized to comply with academic writing standards.

Comments 15: Line 355, don’t use abbreviation to initiate a sentence.

Response 15: Thank for your suggestion. Thank you for your meticulous review of the manuscript’s writing norms. We have revised Line 355 and verified the entire manuscript to avoid similar issues.

Comments 16: Conclusion section, meat quality should be considered in making a conclusion.

Response 16: Thank for your suggestion. We have revised the Conclusion section to incorporate key results related to meat quality.

Comments 17: Several references appeared to have incomplete information. It is recommended that the authors carefully verify and supplement all required bibliographic details to ensure compliance with the journal’s reference formatting guidelines.

Response 17: Thank you for your careful review and important reminder. We have thoroughly checked the formatting of the references and have added the missing information.